# Number Concentration of Gold Nanoparticles in Suspension: SAXS and spICPMS as Traceable Methods Compared to Laboratory Methods

**DOI:** 10.3390/nano9040502

**Published:** 2019-04-01

**Authors:** Alexander Schavkan, Christian Gollwitzer, Raul Garcia-Diez, Michael Krumrey, Caterina Minelli, Dorota Bartczak, Susana Cuello-Nuñez, Heidi Goenaga-Infante, Jenny Rissler, Eva Sjöström, Guillaume B. Baur, Konstantina Vasilatou, Alexander G. Shard

**Affiliations:** 1Physikalisch–Technische Bundesanstalt (PTB), 10587 Berlin, Germany; raul.garcia_diez@helmholtz-berlin.de (R.G.-D.); michael.krumrey@ptb.de (M.K.); 2National Physical Laboratory (NPL), Middlesex TW11 0LW, UK; caterina.minelli@npl.co.uk (C.M.); alex.shard@npl.co.uk (A.G.S.); 3LGC Limited, Teddington TW11 0LY, UK; dorota.bartczak@lgcgroup.com (D.B.); susana.nunez@lgcgroup.com (S.C.-N.); heidi.goenaga-infante@lgcgroup.com (H.G.-I.); 4RISE Research Institutes of Sweden AB (SP), 11428 Stockholm, Sweden; jenny.rissler@ri.se (J.R.); eva.sjostrom@ri.se (E.S.); 5Federal Institute of Metrology (METAS), 3003 Bern-Wabern, Switzerland; baur.guillaume@gmail.com (G.B.B.); konstantina.vasilatou@metas.ch (K.V.)

**Keywords:** nanoparticles, suspensions, traceability, comparison, number concentration, laboratory methods

## Abstract

The industrial exploitation of high value nanoparticles is in need of robust measurement methods to increase the control over product manufacturing and to implement quality assurance. InNanoPart, a European metrology project responded to these needs by developing methods for the measurement of particle size, concentration, agglomeration, surface chemistry and shell thickness. This paper illustrates the advancements this project produced for the traceable measurement of nanoparticle number concentration in liquids through small angle X-ray scattering (SAXS) and single particle inductively coupled plasma mass spectrometry (spICPMS). It also details the validation of a range of laboratory methods, including particle tracking analysis (PTA), dynamic light scattering (DLS), differential centrifugal sedimentation (DCS), ultraviolet visible spectroscopy (UV-vis) and electrospray-differential mobility analysis with a condensation particle counter (ES-DMA-CPC). We used a set of spherical gold nanoparticles with nominal diameters between 10 nm and 100 nm and discuss the results from the various techniques along with the associated uncertainty budgets.

## 1. Introduction

Nanoparticles are increasingly used in innovative products manufactured by advanced industries and provide enhanced, unique properties of great commercial and societal value. The demand for high performance materials places increasingly stringent tolerances on the properties of nanoparticles. Advances in industrial production and research have made it possible to manufacture new kinds of nanoparticles, which open new possibilities for technological progress in, for example, drug delivery, medical imaging and electronics [1,2]. This progress has raised opportunities as nanoparticles are promising candidates for applications in industry, medicine, research and technology [3], but also environmental and health concerns as more people are in direct contact with them [4,5]. Therefore, there is an urgent need for traceable measurements of nanoparticles, for example in terms of size, shape and concentration, which are also needed to comply with recent European regulations in the area of nanomaterials [6,7,8,9,10]. Recent advances in metrological research have led to the development of methods for the accurate measurement of shapes and the traceable determination of sizes of nanoparticles [11,12].

Although some studies on concentration measurements for single methods exist [13,14,15], no comprehensive study on concentration determination has been attempted for the different methods available. Industrial manufacturers often have access to laboratory methods only. Therefore, a study which attempts to comprehend the limits of laboratory methods for measurements of concentration of nanoparticles in suspensions is timely. In this paper, for the first time, we establish traceable and accurate methods for the measurement of colloidal number concentration. The results of measurements of the concentration of colloidal nanoparticles from traceable reference methods are compared to laboratory methods widely available to stakeholders. Additionally, the study identifies a route for the development of much-sought-for relevant particle reference materials which currently do not exist and could be used for validation of the instrument calibration methods.

## 2. Materials and Methods

### 2.1. Materials

Gold nanoparticles with nominal diameters of 5nm, 10nm, 30nm, 100nm, 250nm and 500nm were purchased from BBI International (Cardiff, UK). The samples were treated according to the dilution protocols developed for the InNanoPart project. Details are given in the Appendix B. For all techniques where sample dilution was required, this was performed gravimetrically to produce a more precise dilution factor.

All samples were prepared as aqueous dispersions of spherical particles. The gold particles with a nominal polydispersity p<10% were stabilized with citrate. The nominal concentrations of all prepared suspensions were estimated from the diameter of the particles measured by transmission electron microscopy (TEM) and the mass of gold in the suspension measured following the exact matching bracketing approach with an internal standard. In detail, samples and standards were prepared with an internal standard of Pt so that the signal intensity ratio of Au/Pt was approximately unity. The concentration of analyte (Au) in the sample was calculated according to Equation (Equation 1):(1)$$w_{X}^{\prime }=\lambda \cdot w_{z}\cdot \frac{m_{Y}}{m_{X}}\cdot \frac{m_{Zc}}{m_{Yc}}\cdot \frac{R_{B}^{\prime }}{R_{Bc}^{\prime }}.$$

Here $w_{X}^{\prime }$ is a mass fraction of the analyte in sample *X* obtained from one measurement, $w_{Z}$ is a mass fraction of the analyte in primary standard *Z*, $m_{Y}$ is the mass of internal standard *Y* added to sample *X* to prepare blend B=X+Y, $m_{X}$ is the mass of sample *X* added to internal standard *Y* to prepare blend B=X+Y, $m_{Zc}$ is the mass of the primary standard solution *Z* added to internal standard *Y* to make calibration blend Bc=Y+Z, $m_{Yc}$ is the mass of internal standard *Y* added to the primary standard solution *Z* to make calibration blend Bc, $R_{B}^{\prime }$ is the measured intensity ratio of the analyte and the internal standard in sample blend *B*, $R_{Bc}^{\prime }$ is the measured intensity ratio of the analyte and the internal standard in calibration blend Bc and λ is the blend-to-blend variation. The values obtained by this method are referred to as nominal values.

### 2.2. Methods

Small-angle X-ray scattering (SAXS) and single particle inductively coupled plasma mass spectrometry (spICPMS) are defined as reference methods. Here the dynamic mass flow method of spICPMS was used. Both methods fulfill the requirement as methods of exceptional scientific status which are sufficiently accurate to stand alone in the determination of number concentration for the certification of a reference material. The two methods also have a firm theoretical foundation so that systematic error is negligible relative to the intended use [16]. The laboratory methods investigated were particle tracking analysis (PTA), dynamic light scattering (DLS), which was used as a stand-alone method or with differential centrifugal sedimentation (DCS) to provide the particle size, ultraviolet visible spectroscopy (UV-vis), and electrospray-differential mobility analysis with a condensation particle counter (ES-DMA-CPC). Here, the focus was on a variety of methods, which are accessible and highly represented in industry (e.g., tabletop, easy-to-use, analysis software, etc.). For example, PTA fulfills the definition of a reference method used above, however SAXS and spICPMS are both traceable respective to number concentration directly to a SI unit (metre for SAXS and kilogram for spICPMS), which means that these methods are directly related to the respective SI-unit via an unbroken chain of comparisons, each having a stated uncertainty [11,14,17]. The traceability substantiate a foundation of the methods on nature constants, which is crucial for metrology. None of the laboratory methods is traceable to a SI-unit respective to the number concentration measurements.

#### 2.2.1. Small-angle X-ray Scattering

SAXS measurements were performed at the four-crystal monochromator beamline (FCM) of PTB [18] with the SAXS setup at the BESSY II synchrotron radiation facility (Helmholtz–Zentrum, Berlin, Germany) at the fixed photon energy E=(8000±8)eV. The samples were filled in vacuum-proof borosilicate glass capillaries with a length of 80.0mm, a width of 4.2mm and a total thickness of 1.25mm. These capillaries produced by Hilgenberg (Malsfeld, Germany) have a wall thickness of ≈120μm. The deviation of the glass wall thickness is less than 2% for a horizontal range of 2.5mm. This range is at least five times larger than the typical beam diameter, avoiding the convolution of different thicknesses in the measurement. The thickness of the sample introduced in the capillary is homogeneous within 2%.

The lower section was filled with Fluorinert FC-3283, acquired from Iolitec (Heilbronn, Germany), which has a density of $\rho =1.82\;g\cdot cm^{-3}$, a low viscosity and is immiscible with aqueous solutions. The sample was filled on top of the Flourinert FC-3283. The capillaries were sealed. With the help of a calibration liquid, the sample thickness *w* was determined in a traceable way. The capillaries were scanned with a step size of 0.5mm to ensure that there was no sedimentation during the experiment, as in this case the concentration at all heights would be the same. At each point, a scattering pattern was obtained with an exposure time of t=60s. The scattering patterns were collected with a Pilatus 1M detector (Dectris Ltd., Baden, Switzerland) with a pixel size of $s\cdot s=(172.1\pm 0.2)\cdot (172.1\pm 0.2)\;\mu m^{2}$ [19]. The measured intensity $I_{meas}\left(q\right)$ had to be normalized to the quantum efficiency $\eta_{QE}$ of the Pilatus detector at the given energy of 8000eV. The transmitted photon flux Φ was measured with and without a sample to access the sample transmittance $T_{sample}$ to calculate the sample thickness *w*. The incoming flux $\Phi_{0}$ was measured by the calibrated silicon diode. The distance between the detector and the sample was set to L=(4511±5)mm. All scattering curves obtained from the data were normalized to *T*, which is a total transmission of the sample and the capillary, $\Phi_{0}$ and *t*. The scattering from the particles is proportional to the square of the contrast $\Delta \rho =\rho_{e\_NP}-\rho_{e\_slv}$, where $\rho_{e\_NP}$ is the electron density of the nanoparticles and $\rho_{e\_slv}$ is the solvent electron density.

For the calculation of the contrast, a gold density of $\rho_{NP}=\left(19.30\pm 0.02\right)\;g/cm^{3}$ was used [20,21,22]. A detailed explanation is provided in the Section A.1.

#### 2.2.2. Single Particle Inductively Coupled Plasma Mass Spectrometry

SpICPMS measured number concentrations of inorganic particles using an Agilent 8900 ICPMS/MS instrument (Agilent, Santa Clara, CA, USA). The instrument used the MassHunter4.3 software (Agilent) and is capable of microsecond detection, which makes the analysis in single particle mode possible. For analyses in fast transient analysis (TRA), the newly developed single particle application module of the ICPMS MassHunter software (G5714A) was used. For these analyses, a dwell time of 100s per point and “no gas” setting was used. All analyses were performed at fixed total acquisition time of 60s.

The transport efficiency η was determined using the dynamic mass flow method. For each measurement the steady state was ensured before the start of the run. After stabilization, the weight of solution going into the instrument (sample uptake) was recorded every 3 min over 15 min, whilst the weight of solution going into the nebulizer (sample uptake minus waste measured directly) was recorded over 45 min, every 5 min. Recorded weights were plotted versus the time at which the weights were recorded. Both the mass of solution going into the nebulizer and the total uptaken mass (going into the instrument) are represented as slopes of the linear regression. The transport efficiency η is calculated as follows from the nebulizer slope and the sample uptake slope:(2)η=(nebulizerslope)/(sampleslope).
The particle concentration (*C*) in the sample was derived from Equation (Equation 3):(3)$$C=\left(N\cdot D_{f}\cdot 1000\right)/\left(\eta \cdot V_{m}\right),$$
where *N* is the number of particles detected in the time scan, $D_{f}$ is the sample dilution factor of the samples and $V_{m}$ is the sample mass flow [17]. The results of additional spICPMS measurements, performed with different setup at RIKILT Wageningen UR (DLO), are listed in Section C.1.

#### 2.2.3. Particle Tracking Analysis

All the PTA measurements and the analysis of the video files were performed with an NS500 (Malvern Instruments, Malvern, UK) equipped with a violet diode laser as a light source (405nm CW, max power $P_{max}<60\;\mathrm{mW}$), an EMCCD camera and NTA3.2 software (Malvern).

The temperature was set and maintained at $\left(25\pm 1\right){\;}^{{}^{\circ}}C$ throughout the measurements. Each aliquot was measured at least five times under the same conditions with a fresh portion of the sample loaded before each 60s long measurement. All the samples were measured in triplicate on three different days (total of nine aliquots per sample and 45 individual videos). The detection threshold was set by the operator and the associated uncertainty was included in the final budget. The method trueness was estimated with 8013 NIST reference gold nanoparticles with 60nm nominal diameter and also included in the final uncertainty budget.

Prior to the analysis, the as-received materials were prepared and diluted in a 1mM trisodium citrate buffer. All dilutions were prepared gravimetrically. Alongside the InNanoPart samples, the sample of gold nanoparticles (8013 NIST) was used as a quality control for gold samples. With help of 8013 NIST sample the trueness of the method was tested to calculate the bias in particle counting with PTA. No bias was found, but the method recovery ($R_{m}=1.0\;$) and the uncertainty component associated with the bias in particle counting with PTA were calculated based on 8013 NIST and included in the overall uncertainty budget.

To calculate the measurement uncertainty, concentration values obtained from five replicate measurements of each aliquot were averaged first, as recommended by the instrument’s manufacturer. This gave a total number of nine independent size values (three independently prepared aliquots measured on three different days). Then, using an Analysis of variance (ANOVA) table, the day-to-day reproducibility and the repeatability within one day were calculated. The uncertainty associated with the detection threshold parameter set by the user was estimated experimentally by varying this parameter over a reasonable range and looking at the effect this had on the measured particle number. Regression analysis gave the gradient directly. The overall budget also included the uncertainty associated with weighing for the preparation of sample dilutions. The results provided by PTA represent the particle number concentration in the original samples, derived by multiplying the values obtained with the NTA3.2 analysis software and the dilution factor.

#### 2.2.4. Dynamic Light Scattering

The instrument used in the measurements is a Zetasizer Nano from Malvern Instruments using a He–Ne laser (λ=633nm) and measuring the scattered light in a fixed angle of 173°.

The main purpose of DLS is to measure the size of particles from the diffusional motion of the particles. The success of the technique is based on the fact that it can measure size in a wide concentration range with high sensitivity and without the need for calibration. It provides the hydrodynamic diameter and polydispersity index of a particle population in a rapid and user-friendly manner.

The flexibility in the concentration range results from the instrument being able to vary the attenuator and the measurement position in the sample container. The measurements were made in accordance with standards, to the extent that they were applicable for the purpose [23]. The software also provides a measure of the scattering intensity after compensating for the signal attenuation and measurement position in the sample container (derived count rate). The scattering intensity is related to the particle diameter and shape, the refractive index of the particles, the absorption of the particles, calibrated instrument variations and concentration.

Thus, knowing the other parameters, the volume concentration can be derived and is given by the software (estimated from the parameters measured by the DLS) or given as input to the program. The instrument scattering intensity was calibrated with measurements of pure toluene at several timepoints during the period of the measurements and was not observed to change significantly (relative standard deviation, $s_{rel}\approx 4\%$).

For the DLS measurements, the samples had to be diluted to avoid the multiple scattering events and uncertainty introduced by high attenuation. Each sample was measured at several dilution ratios, resulting in concentrations and size estimates at several concentrations and attenuators. The results presented are an average of the results from the concentrations where high quality DLS data was achieved according to the criteria and considerations given below. The typical derived count rate in high quality data was in a range between 1000 and 25,000kcps. The derived count rate is a relative measure of the probability of the colloidal solution to scatter light. In this interval, the effect of multiple scattering is expected to be minimal. However, for some samples, the dilution resulted in a low quality of the size data. Since particle size is crucial when converting scattering intensity (proportional to $d^{6}$) to volume concentrations (proportional to $d^{3}$)—or number concentration—an error in size and/or in the broadening of the distributions will result in a large error in the estimation of concentrations. Thus, the upper concentration range that can be used is limited by multiple scattering events and uncertainties in the attenuation, and the lower range by the quality of the size data (noise, broader distributions, aggregation, etc).

The estimation of concentration (volume or number) is very dependent on the refractive index *n* and the absorption *A* inserted into the software. For the calculations, labelled values n=0.183 and A=3.43 were used [24]. Even if the conditions for the measurement of scattering intensity were optimized by diluting the samples, another large uncertainty was identified and that was the size distribution. The measured distributions gained by DLS were much broader than those from the reference method (SAXS). This results in shifts in the estimated volume and number size distributions (*N*) compared to the intensity size distribution (*I*)— which is closest to what the DLS measures.

Since the software does not provide number concentrations and as the broad distributions result in large uncertainties when estimating number concentrations from the intensity, we conclude that the DLS is not the method of first choice for estimating number concentrations. The resulting size, polydispersity index and volume concentrations are given in tables using the “normal resolution” algorithm as well as the “high resolution” algorithm (see Section C.2) [25,26].

#### 2.2.5. Ultraviolet Visible Spectroscopy

Spectra for UV-vis were acquired in quartz cuvettes using a LAMBDA 850 spectrophotometer (Perkin Elmer Inc., Waltham, MA, USA). Samples were analyzed over the wavelength range λ=250nm to λ=800nm. The number concentration *N* of the particles was derived from the absorption of the particles according to the formula:(4)$$N=\left(2.303\cdot A_{405}\right)/\left(L\cdot C_{405}\right),$$
where $A_{405}$ is the adsorption measured at λ=405nm, *L* is the optical path and $C_{405}$ is the particle extinction cross section calculated using Mie theory with a refractive index *n* and extinction coefficients of 1.62 and 1.95 respectively for gold and 1.357 and 0 for water. The size of the particles used for the calculation of the extinction cross section was measured by TEM, where the particles were modeled as a sphere. The results acquired with the UV-vis method are shown in the Section C.3.

#### 2.2.6. Differential Centrifugation Sedimentation

The DCS technique measures the time of sedimentation of particles through a liquid medium, exposed to a centrifugal field, where the sedimentation time is a function of the particle size and density. The Stokes’ diameter of the particles is determined based on a spherical approximation, assuming knowledge of the mean density of particles. Because the average density of the Stokes’ volume of the particles was unknown, the DCS results were combined with the particle hydrodynamic diameters measured by DLS and analyzed with the cumulant method (see appendix). Details for the DLS methods used for DCS measurements are given in the Section A.3 For a sphere, the Stokes’ volume and the hydrodynamic volumes are in fact equivalent.

DCS was performed using a CPS 24,000 disc centrifuge (CPS Instruments Inc., Stuart, FL, USA) equipped with an LED laser emitting light with a wavelength between λ=385nm and λ=425nm and with a spectral intensity peak at λ=405nm. The instrument was operated at ω = 20,000 rpm with a typical 14mL
8% to 24% (*w*/*w*) sucrose gradient in water (average gradient density $\rho_{f}=1.064\;g\cdot cm^{-3}$) if not otherwise stated. This was generated by injections of decreasing sucrose concentration, followed by a final addition of 0.5mL dodecane as an evaporation barrier. A period of 30 min was allowed prior to the measurement to facilitate thermal equilibrium. A calibration of the instrument was performed before each sample injection by using either polyvinyl alcohol (PVA) or polystyrene (PS) calibration particles provided by the instrument manufacturer with a nominal diameter of d=237nm and d=522nm and a density $\rho =1.385\;g\cdot cm^{-3}$ and $\rho =1.052\;g\cdot cm^{-3}$. The injected sample volume was measured by weighing the syringe containing the sample before and after each injection and assuming a density of the sample solution of $\rho =1\;g\cdot cm^{-3}$.

The particle number concentration was measured by integrating the measured weight-based size distribution and dividing the measured mass by the average mass of a single particle. The results acquired with the DCS method are shown in the Section C.4 as acquired.

#### 2.2.7. Electrospray-Differential Mobility Analysis with Condensation Particle Counter

The ES-DMA-CPC measurements were carried out using an experimental setup consisting of a pressure-based flow controller (MFCS EZ pressure generator, Fluigent, Jena, Germany), a liquid flow meter (SLG-0075, Sensirion, Staefa, Switzerland), a three-way valve, a mass flow meter (MFM, red-y smart meter GSM, Vögtlin Instruments, Aesch, Switzerland), an ES source (model 3482, TSI Inc., Shoreview, MN, USA), a tube furnace (model FRH-25/150/1100, Linn High Therm, Hirschbach, Germany), a DMA (model 3080, TSI Inc.) and a CPC (model 3776, TSI Inc.).

A lab-built sample injection system based on a pressure flow controller was used to deliver the suspension to the electrospray generator. The nanoparticle (NP) suspension was injected into the ES through a 25μm or 40μm diameter fused-silica capillary and aerosolized in the ES chamber due to the high negative voltage applied on the orifice plate. The suspension flow rate was monitored with the use of a liquid mass flow meter (MFM). Compressed air was used as a carrier gas and the flow rate was recorded continuously with the MFM. The highly charged aerosol resulting from nebulization in the ES generator was then neutralized in the soft X-ray chamber of the ES and guided through the tube furnace operated in the temperature range $\vartheta =20{\;}^{{}^{\circ}}C-500{\;}^{{}^{\circ}}C$. The dimensions of the tube, and hence the residence time of the particles in the furnace, were optimized such that interferences due to residue particles could be reduced [27], while losses due to heating and particle diffusion remained negligible. To determine the resulting particle size distribution and number concentration, the aerosol passed through a DMA operated in a continuous mode, where the particles were classified in an applied electric field according to their electrical mobility and counted by the CPC at the DMA exit.

By measuring the suspension flow rate through the capillary $Q_{liq}$, the carrier gas flow rate $Q_{air}$, and the concentration of the NPs in the aerosol phase $C_{gas}$, the concentration of the NPs in the suspension, $C_{liq}$, can be calculated using the following quation:(5)$$C_{liq}=\left(Q_{air}\cdot C_{gas}\right)/Q_{liq}.$$

The number concentrations of NPs in the suspensions $C_{liq}$, measured with the ES-DMA-CPC method, and the corresponding standard measurement uncertainties are summarized in Section 3 and shown as acquired in Section C.5. It is known that particle losses occur in the electrospray upon aerosolization of the suspension, however, these losses cannot be quantified unless the method is calibrated with reference suspensions [28]. Our results have therefore not been corrected for possible particle losses in the electrospray. The electrospray transmission efficiency (and hence the measurement efficiency of the whole setup) increases with decreasing suspension flow rate through the capillary. The values of $C_{liq}$ correspond to the values obtained at the lowest suspension flow rate of $Q_{liq,min}$ that could be attained experimentally.

## 3. Comparison of the Results and Discussion

The 5nm particles could not be accessed by reference methods and only by few laboratory methods. The 250nm as well as 500nm particles showed very strong sedimentation and therefore it was decided to take into account only 10nm, 30nm and 100nm particles. Despite the inability of some methods to measure all chosen samples, all samples were measured by at least five different techniques. With the SAXS setup used in this work the number concentration the large particles could not be determined due to the measurement times being longer than concentration changes due to sedimentation. For spICPMS, a deconvolution algorithm was proposed, which can help to access Au10 particles [29]. However, as such approach is likely to introduce an additional uncertainty, it was considered unsuitable in the context of this work.

Several of the investigated methods provided the mean diameter of the particles which has a strong influence on the subsequent number concentration determination. The measured diameters are given under assumption of the spherical shape of the particles and are shown in Table 1 and in Figure 1. The spISPMS results were provided by particle frequency method, which is not traceable to a SI-unit and delivered standard deviation only. Nominal diameter was determined with TEM. For DLS number size distributions were used.

In Figure 1, results for number weighted mean diameters are shown. SAXS, spICPMS and TEM measure the inorganic core of the gold particles without the coatings used for stabilization. Therefore, these three methods can be compared directly. For the Au30 and Au100 samples, these methods agreed within their uncertainties. For the Au10 sample only SAXS and TEM delivered results, with TEM-value differing by a factor of ≈1.2 from the value measured by SAXS. For TEM, the deviation of diameters from the values measured by SAXS decreases with increasing particle size.

PTA and DLS measure the hydrodynamic radius of the particles, which includes the particle itself, the ligand layer and the associated hydration layer. Therefore they can not be directly compared to other three methods. Nevertheless, they provide useful information about performance of the methods. The PTA method agreed within uncertainties with DLS for the Au30 sample. For the Au100 sample the PTA deviated by a factor of ≈1.27 from the DLS. It has to be mentioned that despite different measurands, DLS agreed within uncertainties with all other methods for the Au10 and Au30 samples, which is a consequence of large uncertainties of the method. However PTA, which provides much lower uncertainties than DLS, agreed within uncertainties with all other methods for the Au100 sample. For DLS, the use of intensity size distribution delivers values for diameters up to 28% larger compared to the results obtained by use of the number size distribution. A comparison of the number-based size distribution measured by SAXS showed that the DLS size distribution measurement broadening is due to instrumental broadening. Because of this, the number-based size distributions measured by SAXS were used to estimate the number concentration by DLS in the next step.

### Number Concentration

The number concentrations with corresponding standard uncertainty u(k=1) acquired with reference and laboratory methods are shown in Table 2 and Figure 2. SAXS and spICPMS measurements were used as reference measurements for colloidal number concentration, which the laboratory methods presented were compared with. Their properties qualified SAXS and spICPMS as reference methods which have the potential to provide internationally recognized standardization of number concentration measurements due to their traceability to the SI-units.

The comparison highlighted the strengths and limitations of the two reference methods. SAXS is highly accurate, but its accuracy is highly reliant on the exact knowledge of the difference of the electron density of the nanoparticles and the solvent. The second limitation was the time needed for a measurement—suspensions of gold particles with a diameter d> 100 nm showed tremendous sedimentation after short time and therefore no reliable number concentrations could be provided by the SAXS setup used in this work.

The spICPMS method could not access the number concentration of the Au10 sample, as part of the particle population was difficult to resolve from the background signal. The Au10 sample could only be measured with SAXS as a reference method, but with the relative standard uncertainty of 16%. This is due to the upper limit of the accessible *q*-range in the setup. For the Au30 sample, SAXS provided accurate values with a relative standard uncertainty of u≤7%. An example of uncertainty budget for the Au30 sample is provided in Table 3.

The number-based concentration of gold nanoparticles was measured with spICPMS following the dynamic mass flow method, which achieved the standard measurement uncertainty (k=1) of less than 8%. A full uncertainty budget associated with the measurement was provided for gold samples in the size range from 30nm to 100nm. The obtained values agreed well (within their uncertainty) with the nominal values, as well as with the values provided by SAXS. The uncertainty contributions (% of total) for spICPMS were estimated in accordance with ISO 17025 and Eurachem/CITAC and are shown exemplary for the Au30 sample in Table 4. The spICPMS method was the only reference method, which could measure the number concentration of Au100 sample. The relative uncertainty of the measurement on this sample was u<7%.

From Table 2 as well as Figure 2a,b it is visible that, for Au10 and Au30 samples, with the exception of the ES-DMA-CPC method, all the methods are in close agreement. All laboratory methods agree with the result provided by SAXS for the Au10 sample. Only ES-DMA-CPC deviates for this sample by a factor of eight. The sample measured by all available methods was the Au30 sample. With the exception of the ES-DMA-CPC method, which differed by a factor of six, all other methods provided results which agreed within their uncertainties. The DCS/DLS measurement yielded a value ≈ 11% lower than SAXS, spICPMS, PTA and DLS, but still within the range of the uncertainties.

Whereas ES-DMA-CPC and SAXS could not provide any results on Au100 samples, the results from spICPMS, PTA and UV-vis as well as the nominal values agree within their uncertainties as shown in Figure 2. The DLS value is higher and the DCS value is lower with the maximal deviation by a factor of two.

With PTA being a light-scattering-based technique, the size limit of detection was found, as expected, to be dependent on the type and density of the material the analyte is composed of. It was possible to detect gold particles as small as 10nm, but not to measure their concentration reliably. Particles of around 30nm and 100nm were suitable for the technique. Compared to the reference methods, PTA delivered values within uncertainties for both accessible samples. An example of the uncertainty budget for PTA and the Au30 sample is shown in the Section A.2.

The estimation of volume or number concentration from the DLS is very dependent on the refractive index *n* and the absorption coefficient *A* of the nanoparticles inserted into the software. For the calculations, the refractive index n=0.183 and the absorption coefficient A=3.43 were used [24]. The uncertainties for volume concentration measurements are much lower than for number concentration estimation. However, even if the conditions for the measurement of the scattering intensity were optimized by diluting the samples, another large uncertainty was identified: the size distribution. The distributions measured by DLS, are much broader than those from the reference method (SAXS). This fact results in shifts in the estimated volume and number size distributions compared to the “intensity” size distribution, which is what the DLS measures (scattering intensity).

The DCS method could provide number concentration for all samples, even though the size had to be measured with DLS. If the diameter of the particles measured with the reference method SAXS was used, the obtained results were in better agreement with the values of the reference methods. As for DLS, DCS relies on the exact knowledge of the refractive index *n* of the nanoparticles for the estimation of the concentration. Thus, for the DCS (DCS/DLS) method, precise knowledge of the particle size and size distribution as well as the density of the material is essential for use in laboratory conditions. The major contribution to DCS uncertainty is composed of the uncertainties on the size and density of the calibrants [30]. Method trueness for the measurement of nanoparticle concentration was found to be around 30%. The total uncertainty is therefore u≥10%.

The UV-vis method could provide reliable number concentration values as only high-density particles were used in the experiments presented, however this methods needs additional size measurements e.g., by TEM, in order to calculate the extinction cross-section. The estimation of the uncertainty for the UV-vis method relies on a comparison to gravimetric methods and theory [31,32,33]. This provides an estimated relative uncertainty for UV-vis of u≤20%.

The electrospray transmission efficiency in ES-DMA-CPC experiments depends on the suspension flow rate through the capillary and the suspension matrix. Compared to the primary methods, the deviation of the ES-DMA-CPC values was up to a factor of eight. Additionally, ES-DMA-CPC could only provide number concentrations for particles with a diameter of d<100nm [28]. However, the results presented are based on the assumption of 100% transport efficiency. The results from this comparison can be used to determine realistic efficiencies for ES-DMA-CPC.

## 4. Summary

The particles smaller than 10nm had to be discarded as only a few laboratory and none of the reference methods could access them in current setup. Gold particles with diameters larger than 100nm showed very strong sedimentation and had to be discarded therefore as well. In this study, SAXS and spICPMS were used as reference methods. However, both methods only provided results for Au30 samples, where both methods agreed within their uncertainties, which showed respective relative uncertainties better than 10% in this case. Both—SAXS and spICPMS—can be recommended as stand-alone methods for simultaneous diameter and number concentration measurements of gold nanoparticles. However, for spICPMS the size can be derived from mass, which is based on too many assumptions for spICPMS to be the method of choice for diameter measurements. SAXS provides additionally the size distribution of the particles, if the particles are sufficiently monodisperse. As alternative DCS can be used to measure the particle size distribution to determine if the batch is sufficiently monodisperse. In addition, multimodal size distributions can be identified by methods with sufficient size resolution, such as DCS, PTA and spICPMS, which are much cheaper than TEM and can be used to measure the particle size distribution for any polydispersity.

The number concentrations measured by PTA agree within the stated uncertainty with reference methods for both measured samples. Under the current conditions, UV-vis delivered results comparable with reference methods for all the measured samples, but with higher uncertainties. Notably, for this type of sample this is the most commonly used and cost-effective method, provided an additional method like TEM delivers the diameters of the particles. DCS provided much higher uncertainties in number concentration measurements and somewhat variable results. It showed good agreement with reference values for smaller particles. For Au100, DCS delivered values up to a factor of 2 lower than the reference values. This deviation could possibly be decreased if the size of the particles is known very well, which makes additional methods like TEM, DLS or SAXS useful as supplementary methods for DCS. The ES-DMA-CPC method needs a reference concentration to calculate the recovery efficiency and is therefore only suited for number concentration evaluation if a reliable reference sample is available or, alternatively, the method proposed in [34] must be used. DLS, as a stand-alone method, has the largest uncertainties and showed the strongest deviation from the reference methods and therefore cannot be recommended for accurate number concentration measurements. For non-dissolving particles with high density, which is close to the bulk density, the nominal concentrations measured with total gold content using the exact matching bracketing ICPMS method and TEM diameters agreed within uncertainties with reference methods.

In general, it could be shown that reference methods agree on number concentration values within their uncertainties, which are better than 10%. Additionally, for all measured samples, agreement of the nominal method with the reference methods could be verified. Furthermore, the chosen laboratory methods showed a good degree of agreement with reference methods and nominal values, even though they provided much higher uncertainties.

## Figures and Tables

**Figure 1 nanomaterials-09-00502-f001:**
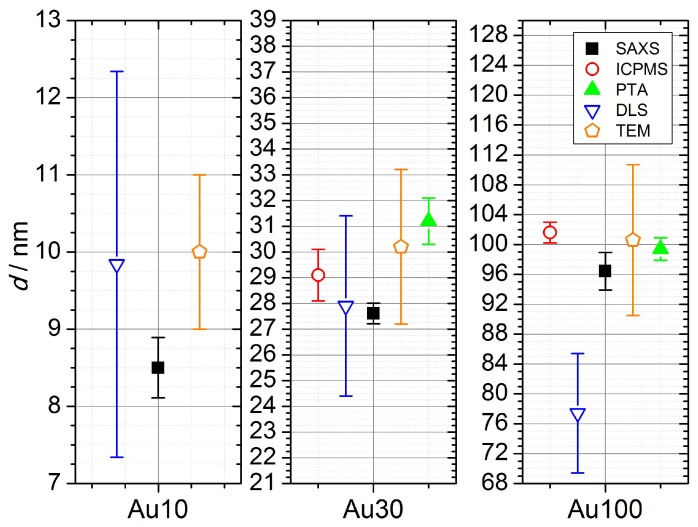
Comparison of the diameters and corresponding uncertainties for reference and laboratory methods. Open symbols symbolize methods without full uncertainty budget for diameter estimation.

**Figure 2 nanomaterials-09-00502-f002:**
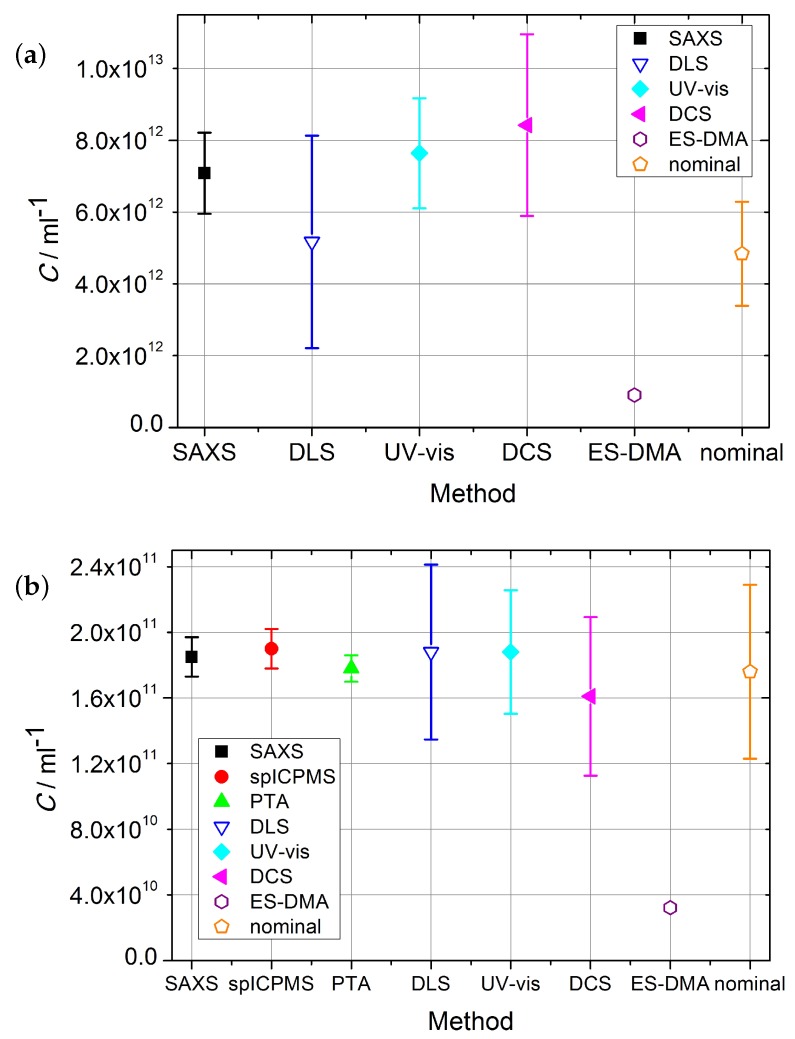
Comparison of number concentrations and corresponding standard uncertainties acquired with reference and laboratory methods for (**a**) Au10, (**b**) Au30 and (**c**) Au100 samples. Open symbols depict methods without full uncertainty budget.

**Table 1 nanomaterials-09-00502-t001:** Number weighted mean diameters of samples. Above the double line results acquired by methods measuring the inorganic core are shown. Below this line results obtained by techniques with access to hydrodynamic radius are depicted.

Method	$d_{\mathrm{Au}10}$/nm	$d_{\mathrm{Au}30}$/nm	$d_{\mathrm{Au}100}$/nm
**SAXS**	8.5±0.4	27.6±0.4	96.4±2.5
**spICPMS**	-	29.1±1.0	101.6±1.4
**TEM**	10.0±1.0	30.2±3.0	100.6±10.1
**PTA**	-	31.2±0.9	99.4±1.5
**DLS**	9.9±2.5	27.9±3.5	78.0±8.0

**Table 2 nanomaterials-09-00502-t002:** Number concentration of the gold suspensions. The tag “(ref)” denotes reference methods.

Method	$C_{\mathrm{Au}10}$/mL${}^{-1}$	$C_{\mathrm{Au}30}$/mL${}^{-1}$	$C_{\mathrm{Au}100}$/mL${}^{-1}$
**SAXS** (ref)	$\left(7.08\pm 1.13\right)\cdot 10^{12}$	$\left(1.85\pm 0.13\right)\cdot 10^{11}$	-
**spICPMS** (ref)	-	$\left(1.80\pm 0.14\right)\cdot 10^{11}$	$\left(4.10\pm 0.26\right)\cdot 10^{9}$
**PTA**	-	$\left(1.78\pm 0.08\right)\cdot 10^{11}$	$\left(4.31\pm 0.24\right)\cdot 10^{9}$
**DLS**	$\left(5.17\pm 2.96\right)\cdot 10^{12}$	$\left(1.88\pm 0.61\right)\cdot 10^{11}$	$\left(8.47\pm 2.18\right)\cdot 10^{9}$
**UV-vis**	$\left(7.64\pm 1.53\right)\cdot 10^{12}$	$\left(1.88\pm 0.38\right)\cdot 10^{11}$	$\left(4.27\pm 0.85\right)\cdot 10^{9}$
**DCS/DLS**	$\left(8.42\pm 2.53\right)\cdot 10^{12}$	$\left(1.61\pm 0.48\right)\cdot 10^{11}$	$\left(2.08\pm 0.62\right)\cdot 10^{9}$
**ES-DMA-CPC**	$\left(9.03\pm 0.32\right)\cdot 10^{11}$	$\left(3.22\pm 0.12\right)\cdot 10^{10}$	–
**Nominal**	$\left(4.84\pm 1.45\right)\cdot 10^{12}$	$\left(1.76\pm 0.53\right)\cdot 10^{11}$	$\left(4.16\pm 1.25\right)\cdot 10^{9}$

**Table 3 nanomaterials-09-00502-t003:** Uncertainty budget for small angle X-ray scattering (SAXS) for the Au30 sample.

Input $x_{i}$	$x_{i}$ · Unit	$u(x_{i})$ · Unit	$u_{C}$/mL${}^{-1}$
*s*	172.1μm	0.2μm	$6.1\cdot 10^{9}\;$
*L*	4511mm	5mm	$5.7\cdot 10^{9}\;$
$\Phi_{0}$	$6.8\cdot 10^{9}\;\mathrm{ph}/s$	$6.8\cdot 10^{7}\;\mathrm{ph}/s$	$1.85\cdot 10^{9}\;$
*T*	2.00%	0.02%	$1.85\cdot 10^{9}\;$
*E*	8000.0eV	0.8eV	$5.6\cdot 10^{8}\;$
*t*	60.00s	<0.06s	$1.85\cdot 10^{8}\;$
$\eta_{QE}$	97%	3%	$6.0\cdot 10^{7}\;$
Nfit	$2.66\cdot 10^{-5}\;$	$1.7\cdot 10^{-6}\;\;$	$9.58\cdot 10^{9}\;$
*w*	1.01mm	0.03mm	$5.46\cdot 10^{9}\;$
${\left(\Delta \rho \right)}^{2}$	$1.88\cdot 10^{7}\;{\mathrm{nm}}^{-6}$	$4.05\cdot 10^{4}\;{\mathrm{nm}}^{-6}$	$4.25\cdot 10^{8}\;$
*C*	$1.85\cdot 10^{11}\;{\mathrm{ml}}^{-1}$		$1.3\cdot 10^{10}\;$

**Table 4 nanomaterials-09-00502-t004:** Relative uncertainty contributions for single particle inductively coupled plasma mass spectrometry (spICPMS) for the Au30 sample.

Sample	Uncertainty Contribution (%)			
	N	$D_{f}$	η	$V_{m}$
Au30	74.1	14.1	11	0.8

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
