# Peer review of "Number Concentration of Gold Nanoparticles in Suspension: SAXS and spICPMS as Traceable Methods Compared to Laboratory Methods"

_nanomaterials, 2019, doi:10.3390/nano9040502_

Round 1

Reviewer 1 Report

The manuscript "Number concentration of nanoparticles in suspension: SAXS and spICPMS as traceable methods compared to laboratory methods", authors Schavkan A. et al. describes recent results obtained in the frame of an european project devoted to develop traceable measurement and calibration protocols to measure particle number concentrations in liquid suspension. The authors compared different methods to investigate the concentration of monodispersed Au nanoparticles of  10, 30 and 100 nm in diameter. The manuscript is in principle interesting particularly for industry in view to identify new approaches based on laboratory methods for the measurement of colloidal number concentration. However, the are some aspects to be revised before publication in Nanomaterials journal:

- Table 1. The values of particle size obtained by PTA and DLS methods can not be appropriately compared with those of the other used techniques. As correctly reported by the authors, both these methods measured the hydrodynamic diameter of the particles and not only the Au NPs core (as for the other reported methods). For this reason, the comments reported below Table 1 on the comparison between PTA/DLS and other techniques must be removed. In this view, the values of diameter obtained by PTA and DLS methods for the different Au NPs can be completely removed from Table 1.

-   Table 2. It can be very useful for the reader indicate in the table the methods used as reference measurements (e.g. including a footnote to the table or other).

- Concerning the number concentration measurements obtained by UV-Vis, the authors must include a comment in the the results and discussion section reporting that in order to appropriately evaluate the concentration TEM measurements are needed (as described in the experimental section).

- In Appendix C Au NPs having an average diameters form 5 to 500 nm. The authors must insert a comment in the results and discussion section the reason why Au Nps below 10 nm and above 100 nm were not reported in the manuscript. 

- Finally, the authors must include in the conclusions a brief comment regarding: a) Which is the best combination of methods in order to evaluate size and number NPs concentration which was identified by the obtained results; b) These methods are reliable also in presence of not monodispersed NPs? How to know if the particles are monodispersed without as example TEM analysis?

Minor:

Page 8 line 302. The sentence "[...] difference of the electron density...[...]" is reported twice.

Author Response

Dear Sir or Madame,

Thank you for the thorough reading. Most of the suggestions/comments were helpful and will make the manuscript more comprehensive.

In the pdf-file the changes are shown with colors:

red - means the text is deleted from the paper

blue - means the text was added to the paper.

In following your comments are shown with brown and my changes in green:

- Table 1. The values of particle size obtained by PTA and DLS methods can not be appropriately compared with those of the other used techniques. As correctly reported by the authors, both these methods measured the hydrodynamic diameter of the particles and not only the Au NPs core (as for the other reported methods). For this reason, the comments reported below Table 1 on the comparison between PTA/DLS and other techniques must be removed. In this view, the values of diameter obtained by PTA and DLS methods for the different Au NPs can be completely removed from Table 1.

Your suggestions are highly appreciated, however diameters provided by PTA and DLS are important for this paper. Strictly speaking e.g. TEM and SAXS are also measuring different values - TEM diameter of every single particle as SAXS measures the ensemble average. Nevertheless, for better understanding the diameters provided by DLS and PTA were moved to a separate section of Table 1 and the discussion was amended.

-   Table 2. It can be very useful for the reader indicate in the table the methods used as reference measurements (e.g. including a footnote to the table or other).

The changes were included as requested

- Concerning the number concentration measurements obtained by UV-Vis, the authors must include a comment in the the results and discussion section reporting that in order to appropriately evaluate the concentration TEM measurements are needed (as described in the experimental section).

The changes were included as requested

- In Appendix C Au NPs having an average diameters form 5 to 500 nm. The authors must insert a comment in the results and discussion section the reason why Au Nps below 10 nm and above 100 nm were not reported in the manuscript. 

The changes were included as requested

- Finally, the authors must include in the conclusions a brief comment regarding: a) Which is the best combination of methods in order to evaluate size and number NPs concentration which was identified by the obtained results; b) These methods are reliable also in presence of not monodispersed NPs? How to know if the particles are monodispersed without as example TEM analysis?

The requested discussion was included in summary.

Best regards,

Alexander Schavkan

Reviewer 2 Report

 This paper is about standard science in nano-particles with many kinds of methods. This is important content for discussing about size and concentration of nano-particles. So it can be published. 

 The below things should be ammended.

p10. l307-310 The sentences are overlapped with l302-306.

Author Response

Dear Sir or Madame,

the changes in the document are indicated with color - red means that the text was removed.

The comment

p10. l307-310 The sentences are overlapped with l302-306.

was changed as requested.

Best regards,

Alexander Schavkan

Reviewer 3 Report

Dear Editor, dear authors,

The manuscript of Alexander Schavkan and co-workers describes the usage of small-angle X-ray scattering (SAXS) and single particle inductively coupled plasma mass spectrometry (spICPMS) as reference techniques to determine size and concentration of gold nanoparticles in solution. They use gold nanoparticles with nominal diameters of 10, 30, and 100 nm and determine, if possible, size and concentration of the gold nanoparticles using SAXS and spICPMS, but also using laboratory methods such as particle tracking analysis (PTA), dynamic light scattering (DLS), differential centrifugal sedimentation (DCS), ultraviolet visible spectroscopy (UV-vis), and electrospray-differential mobility analysis with a condensation particle counter (ES-DMA-CPC). Based on these measurements, the authors calculate the uncertainty budget in the determination of both properties for all methods, thereby allowing for a comparison of the measurement accuracy of all methods. The manuscript is very well written (despite some minor flaws mentioned below) and covers the important topic of how to quantify nanoparticle concentrations, for which (to my knowledge) no consensus has yet been reached within the community. With that I’m highly convinced that the paper will be highly appreciated by the readership reached by Nanomaterials and therefore, I would like to recommend the paper for publication after the MINOR ISSUES (mentioned below) have been corrected.

1. The authors aim to introduce SAXS and spICPMS as reference techniques for the determination of nanoparticle concentrations. The manuscript nicely shows the accuracies that can be obtained when using stable and electron rich nanoparticles, such as gold nanoparticles. There are, however, also other classes of nanoparticles, which either create a much lower electron density contrast and/or are much less stable (e.g., biological nanoparticles such as viruses, exosomes etc.), for which I would expect to observe a lower performance of either SAXS or spICPMS or both. In this respect, the title of the manuscript appears to be too broad and I would suggest to talk about “Number concentration of gold nanoparticles” or metal nanoparticles etc.

2. It would be appreciated if the authors could justify their choice of SAXS and spICPMS to serve as reference methods. They mention that both approaches are traceable to SI units, which seems to be the reason for their choice. I, for example, does not know what “traceable to SI units” implies and why this would be a requirement for being a reference methods. On the other hand, the authors also use PTA, which allows for directly extracting diffusion coefficients from tracking the motion of nanoparticles, i.e., the extracted quantity appears to be also directly traceable to SI units (here µm2/s). It would be good if the authors could specify for the audience, why SAXS and spICPMS are special with respect to the other methods introduced, so that they can be defined as reference methods in order to determine the performance of the so-called laboratory methods.

3. The authors use a bracket notation to indicate the uncertainty. They should clearly define, what this actually means. Comparing for example the entries of Table 1 with Figure 1 shows that 8.5(4) nm has to be understood as (8.5 +/- 0.4) nm, while doing the same with Table 2 and Figure 2 shows that 7.08(113)/mL now means (7.08 +- 1.13)/mL. It is not obvious, at which position the dot has to be placed. Maybe this is a standard notation in metrology, but the work would be easier digestible, if this notation could be clearly indicated at least once.

4. Ref. 23 gives a link to the optical properties of polystyrene but not gold. It would be good to link to the correct entry.

5. The authors use a “normal resolution” and “high resolution” analysis of the DLS data. It should be either specified, how these algorithms work, or links to the corresponding reference should be added.

6. Line 212: n denotes here a rotation frequency, while it refers to refractive indices in the rest of the manuscript. I would suggest to use a different letter for the former.

7. Table 1: For the smallest gold nanoparticles, the SAXS result is much smaller than that of DLS and TEM. Nevertheless, the authors seem to be more convinced in the size determination accuracy of SAXS than of DLS and TEM, as they used the SAXS result for normalization of the diameters (lines 264-266). How can the authors exclude that SAXS underestimates the sizes of the small gold nanoparticles (for example, as they might be deviations between the model used for the analysis of the SAXS data and the real electron density distribution)? Can the authors exclude that the large disagreement between nominal and measured concentration, which is only observed for the “10 nm” gold nanoparticles (Table 2), is due to an underestimation of the nanoparticle size in the SAXS measurement?

8. Table 2: With respect to the uncertainty, PTA seems to perform as well as spICPMS. So what makes spICPMS more suitable as PTA to serve as reference method? Furthermore, lines 137 – 139 suggest that the authors do not report the software-reported concentrations determined using PTA, but corrected this value based on calibration measurements using 8013 NIST. Here it would be interesting to report the correction factor, as many labs routinely use PTA to determine the concentration of biological nanoparticles, but without doing such calibration. It would be interesting to know, how close the software-reported values match to the real concentrations.

9. Lines 286-288: Something went wrong when writing this sentence.

10. Lines 307-310 repeat the content of the lines 303-306.

11. Line 314: The 14% does not match to the corresponding uncertainty in Table 2.

12. Line 474: Is 5 °C correct? Most fridges seem to work at +4 °C?

13. Table A4 and A5: Check the units of the concentrations. There is at least a mistake for Cnum.

Author Response

Dear Sir or Madame,

Thank you for the thorough reading. Your suggestions/comments were helpful and will make the manuscript more comprehensive.

In the pdf-file the changes are shown with colors:

red - means the text is deleted from the paper

blue - means the text was added to the paper.

In following your comments are shown with brown and my changes in green:

1. The authors aim to introduce SAXS and spICPMS as reference techniques for the determination of nanoparticle concentrations. The manuscript nicely shows the accuracies that can be obtained when using stable and electron rich nanoparticles, such as gold nanoparticles. There are, however, also other classes of nanoparticles, which either create a much lower electron density contrast and/or are much less stable (e.g., biological nanoparticles such as viruses, exosomes etc.), for which I would expect to observe a lower performance of either SAXS or spICPMS or both. In this respect, the title of the manuscript appears to be too broad and I would suggest to talk about “Number concentration of gold nanoparticles” or metal nanoparticles etc.

The title of the publication was changed as requested

2. It would be appreciated if the authors could justify their choice of SAXS and spICPMS to serve as reference methods. They mention that both approaches are traceable to SI units, which seems to be the reason for their choice. I, for example, does not know what “traceable to SI units” implies and why this would be a requirement for being a reference methods. On the other hand, the authors also use PTA, which allows for directly extracting diffusion coefficients from tracking the motion of nanoparticles, i.e., the extracted quantity appears to be also directly traceable to SI units (here µm2/s). It would be good if the authors could specify for the audience, why SAXS and spICPMS are special with respect to the other methods introduced, so that they can be defined as reference methods in order to determine the performance of the so-called laboratory methods.

To explain this point additional explanation and discussion as well as citation from UIPAC Gold Book were included

3. The authors use a bracket notation to indicate the uncertainty. They should clearly define, what this actually means. Comparing for example the entries of Table 1 with Figure 1 shows that 8.5(4) nm has to be understood as (8.5 +/- 0.4) nm, while doing the same with Table 2 and Figure 2 shows that 7.08(113)/mL now means (7.08 +- 1.13)/mL. It is not obvious, at which position the dot has to be placed. Maybe this is a standard notation in metrology, but the work would be easier digestible, if this notation could be clearly indicated at least once.

The notation was changed from bracket notation to (8.5 +/- 0.4) nm type.

4. Ref. 23 gives a link to the optical properties of polystyrene but not gold. It would be good to link to the correct entry.

The link was amended as requested

5. The authors use a “normal resolution” and “high resolution” analysis of the DLS data. It should be either specified, how these algorithms work, or links to the corresponding reference should be added.

Additional text as well as citations were provided

6. Line 212: n denotes here a rotation frequency, while it refers to refractive indices in the rest of the manuscript. I would suggest to use a different letter for the former.

n was changed to ω as requested.

7. Table 1: For the smallest gold nanoparticles, the SAXS result is much smaller than that of DLS and TEM. Nevertheless, the authors seem to be more convinced in the size determination accuracy of SAXS than of DLS and TEM, as they used the SAXS result for normalization of the diameters (lines 264-266). How can the authors exclude that SAXS underestimates the sizes of the small gold nanoparticles (for example, as they might be deviations between the model used for the analysis of the SAXS data and the real electron density distribution)? Can the authors exclude that the large disagreement between nominal and measured concentration, which is only observed for the “10 nm” gold nanoparticles (Table 2), is due to an underestimation of the nanoparticle size in the SAXS measurement?

SAXS is not used for comparison anymore. Diameters as measured are shown. Obsolete sentenses and comparison were removed.

8. Table 2: With respect to the uncertainty, PTA seems to perform as well as spICPMS. So what makes spICPMS more suitable as PTA to serve as reference method? Furthermore, lines 137 – 139 suggest that the authors do not report the software-reported concentrations determined using PTA, but corrected this value based on calibration measurements using 8013 NIST. Here it would be interesting to report the correction factor, as many labs routinely use PTA to determine the concentration of biological nanoparticles, but without doing such calibration. It would be interesting to know, how close the software-reported values match to the real concentrations.

Here additional discussion on reference methods was included. According to Gold book of UIPAC PTA could also be a reference method, however the aim of the project was to find a reference material (if possible). Therefore here methods traceable to SI-unit (an unbrocken chain of comparisons) were defined as reference methods. This is important for metrological impact. Respective discussion was included into the text of the paper. The 8013 NIST sample was only used for comparison. No devations were found and therefore no additional factor was applied to the number concentration values measured by PTA.

9. Lines 286-288: Something went wrong when writing this sentence.

The sentence was amended accordingly.

10. Lines 307-310 repeat the content of the lines 303-306.

The repeating lines were delted.

11. Line 314: The 14% does not match to the corresponding uncertainty in Table 2.

The value was amended.

12. Line 474: Is 5 °C correct? Most fridges seem to work at +4 °C?

The temperature was changed to +4 °C, as this is the correct value.

13. Table A4 and A5: Check the units of the concentrations. There is at least a mistake for Cnum.

The units were changed accordingly.

Alexander Schavkan
